# A Fault Diagnosis Approach for Rolling Bearing Based on Convolutional Neural Network and Nuisance Attribute Projection under Various Speed Conditions

**Huijie Ma, Shunming Li * and Zenghui An**

College of Energy & Power Engineering, Nanjing University of Aeronautics and Astronautics, Nanjing 210016, China; mahuijie236@163.com (H.M.); me_anzenghui@163.com (Z.A.)

\* Correspondence: smli@nuaa.edu.cn; Tel.: +86-1360-519-9671

**Abstract:** Intelligent fault diagnosis is a promising tool for processing mechanical big data. It can quickly and efficiently process the collected signals and provide accurate diagnosis results. However, rotating machinery often works under various speed conditions, which makes it difficult to extract fault features. Inspired by speech recognition, the nuisance attribute projection method in speech recognition is introduced into fault diagnosis to solve the problem of feature extraction in variable speed signals. Based on the idea of unsupervised feature learning, the loss function of nuisance attribute projection is added to the loss function of convolutional neural network (CNN) to learn fault features from original data. Health status is classified according to the learned characteristics and projection matrix P. A special designed bearing dataset is employed to verify the effectiveness of the proposed method. The results show that the proposed method has a higher accuracy and a simpler framework, which is superior to the existing methods in bearing fault diagnosis.

**Keywords:** rolling bearing; fault diagnosis; variable speed condition; convolutional neural network; nuisance attribute projection

## 1. Introduction

With the introduction of artificial intelligence method into fault diagnosis, mechanical equipment fault diagnosis has been becoming more intelligent, efficient and some problems can be solved by neural network, rather than complex time-frequency analysis methods [1]. Rolling bearing is an important part which is widely used in mechanical equipment, especially rotating machinery, whose maintenance cost is very high. Thus, it is necessary to design an effective fault diagnosis system to monitor the status of rolling bearings [2]. Abnormal vibration signal will be generated due to the variable compliance effect, manufacturing defects and assembly related problems [3]. These signals can be processed to determine the source of fault, and then repaired and maintained accurately. Due to engineering requirements, rolling bearings are working at different speeds and loads. Furthermore, vibration signals of variable working condition contain abundant state information, and some fault features which are not easy to extract in constant condition may be fully displayed [4]. Therefore, it is significant to extract fault information of rolling bearing under variable working conditions.

With the development of technologies of the sensor, signal processing and engineering measurement, many methods have been developed in recent years, such as wavelet analysis, empirical mode decomposition (EMD), fast kurtogram and so on [5,6]. However, these methods have some limitations in analyzing variable speed signals. Recently, many researchers used order tracking and its improved algorithms to extract fault features in variable speed [7–12]. Resampling the vibration signal

in angle-domain so that the fault characteristics of bearing defects are not affected by the rotating speed. As the most popular algorithm in order tracking, the computational order tracking method has a series of assumptions for the sampling process, which leads to the feature extraction error. In recent years, another widely used method deep neural network (e.g., convolutional neural networks (CNN) [13], stacked autoencoder (SAE) [14], deep belief network (DBN) [15]) has been proved to be advantageous and feasible in fault pattern recognition [16–19]. Different from traditional methods based on physical meaning of signal, deep neural networks construct an end-to-end fault diagnosis system and adaptively extract fault features. However, the neural network does not know which fault features do not change with operation conditions and which are the nuisance features. Therefore, the feature extraction process cannot extract obvious fault information from the original features with mixed operation condition attributes [20].

Nuisance attribute projection (NAP) is a method of eliminating channel interference for solving channel mismatch problem in speech recognition at the very beginning [21]. It was widely used in speaker recognition, face recognition and image recognition [22–24]. Wenyi Huang [25] introduced NAP to rolling bearing fault diagnosis and verified its effectiveness in performance degradation assessment and fault pattern recognition. In fact, NAP minimizes the value of the interference attribute by projecting the feature into another space to increase the proportion of fault feature [26]. However, when solving the projection matrix, it is complex that the minimization problem transformed into solving eigenvalues and eigenvectors problem. In addition, when NAP is applied to fault diagnosis, it has massive diagnostic steps that are not a simple and efficient method. In this paper, based on previous research, a new approach that applies NAP to CNN is put forward, and the objective function of NAP is added to CNN for solution. The proposed method simplifies the NAP solving process and makes the diagnosis steps much clearer and more efficient.

This paper is organized as follows. In Section 2, CNN and NAP are briefly described. Section 3 details the proposed learning method CNN-NAP. In Section 4, the proposed method is applied to the diagnosis of variable speed bearing data sets. In Section 5, the proposed method is compared with some traditional methods. Finally, conclusions are drawn in Section 6.

## 2. Background

This section describes the concept of CNN, NAP and expounds a new method on multi-speed bearing fault diagnosis by combining the two methods above.

### 2.1. Convolutional Neural Network

As a classical and widely used structure of deep learning, CNN overcomes some problems that were difficult to be solved by artificial intelligence in the past. The local connections, shared weights and down sampling operation of CNN can effectively reduce the complexity of the network and the number of training parameters [27–30].

There are three major layers in CNN:

1. convolutional layer,
2. pooling layer,
3. fully connected layer.

The convolution layer applies a certain number of filters to obtain the feature map of the input signal. Concretely, the convolution is to take a dot product between a kernel $\omega_c \in R^m$ and the $j$th segmented signal $s^i_{j-m+1:j} \in R^m$ to obtain features

$$c_j = \text{Relu}(\sum_{i=1}^{n} w_c * s^i_{j-m+1:j} + b_c), \tag{1}$$

where $*$ is convolution operator, $\omega_c$ is referred to as the convolution kernel, $b_c$ is the corresponding bias, $n$ is the number of kernels, and $c_j$ is the $j$th output point of the convolutional layer. Relu is activation function. The pooling layer is the down sampling layer reducing the input feature dimension. In this paper, the Max pooling function is used, and the maximum value within a certain sub-region is returned as follows:

$$p_j = \max\{c_{j\times k:(j+1)\times k}\}, \tag{2}$$

where $k$ is the pooling length, and $p_j$ is the pooling output of the $j$th point. After several alternating convolution layers and pooling layers, a fully connected layer is flattened from the output of the upper layer. Based on fully connected (FC) layer output $f$, the health conditions of machines are estimated in the health condition output layer through the softmax regression,

$$y = \frac{1}{\sum\limits_{i=1}^{K} e^{((w_i)^T f + b)}} \begin{bmatrix} e^{((w_1)^T f + b)} \\ e^{((w_2)^T f + b)} \\ \vdots \\ e^{((w_k)^T f + b)} \end{bmatrix}, \tag{3}$$

where $w_i$ is the weight matrix connecting to the $i$th output neuron, $b$ is the corresponding bias vector, and $K$ denotes the health condition categories.

The typical structure of the classic LeNet-5 CNN model [31] is shown in Figure 1. LeNet-5 was applied to recognize the handwritten digit characters and computer printed characters. There are two convolutions layers, two pooling layers and two fully connected layers in this model. In this research, a modified LeNet-5 CNN is designed to solve the classification task on the fault diagnosis.

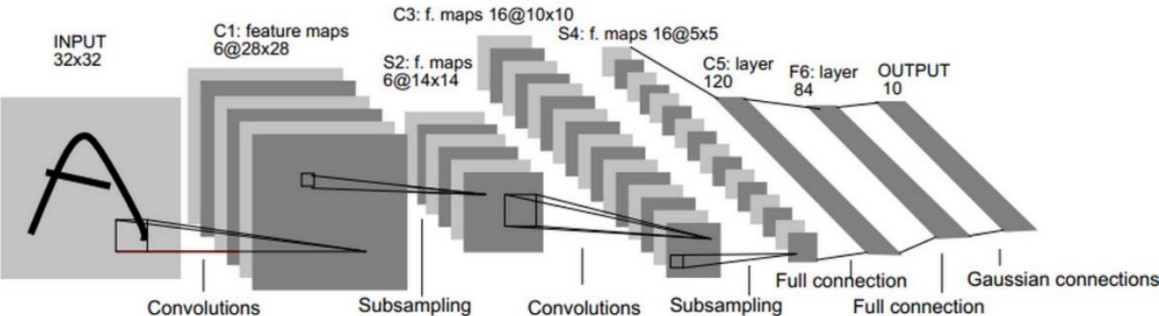

**Figure 1.** The typical structure of LeNet-5 convolutional neural network (CNN) model.

The design CNN structure is shown in Figure 2, which is like a LeNet-5 CNN model. The detail definitions would be discussed in Section 3, and the CNN is developed based on TensorFlow.

### 2.2. Nuisance Attribute Projection

Nuisance attribute projection was initially used in speaker recognition systems to reduce noise and interference information among channels improving the recognition performance of the system. In fact, the NAP algorithm constructs a projection matrix to reduce the distance of different channels, languages and other features under the same speaker in the projection space. Correspondingly, these nuisance attributes are load, rotating speed and noise in fault diagnosis.

Consider a data matrix $F = [f_1, f_2, \ldots, f_n]$ with $n$ vectors, where $f_i$ is feature vectors of $N$-dim. Different $f$ stands for different working conditions. For the same fault, when the feature vectors $f_i$ and $f_j$ of its two working conditions, the distance between them in the feature space is defined as:

$$\delta = \left\| f_i - f_j \right\|^2. \tag{4}$$

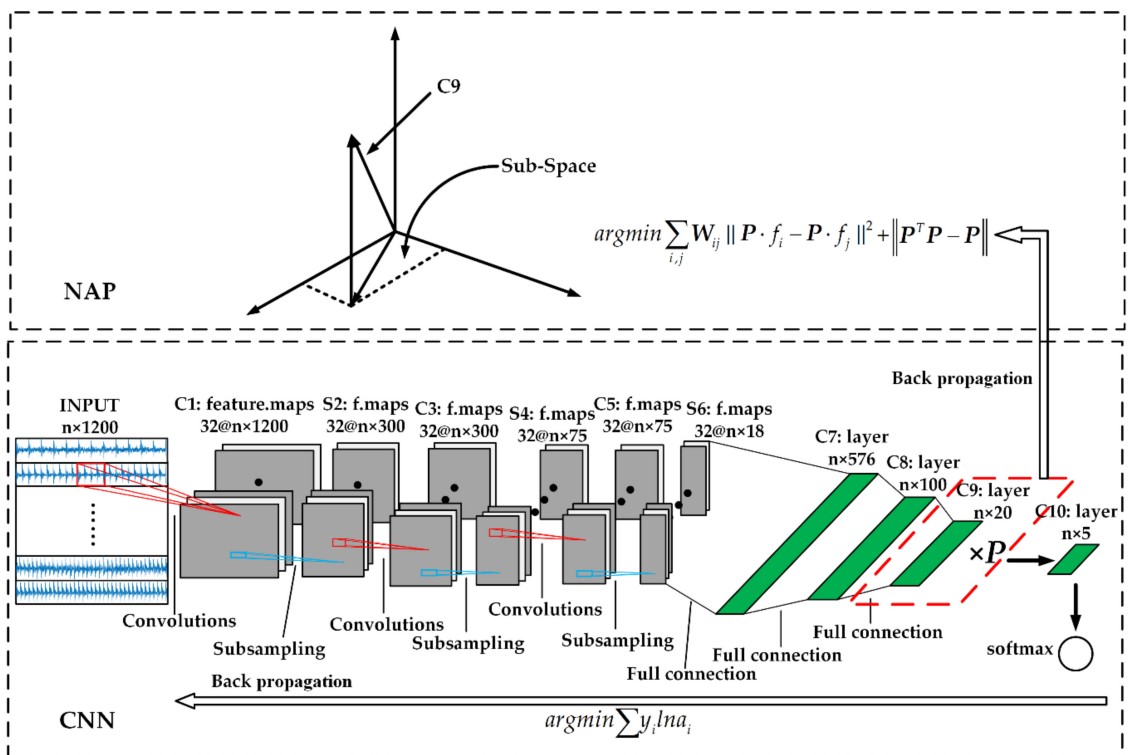

**Figure 2.** The structure of the proposed method.

The subtraction of feature vectors in different working conditions of the same fault can eliminate the same information and retain the different information. In order to weaken the different information between working conditions, the high-dimensional matrix $P$ of $N \times N$ is defined to minimize $\delta$:

$$\delta_P = \|P \cdot f_i - P \cdot f_j\|_2. \tag{5}$$

For multiple signals of the same fault, their working conditions may be the same or different. When these signals are in the same condition, it is not necessary to extract the different information. While their conditions are different, it is necessary. Therefore, define the weight matrix $W$, where the matrix element is

$$W_{ij} = \begin{cases} 1 & workingcondition(f_i) \neq workingcondition(f_j) \\ 0 & workingcondition(f_i) = workingcondition(f_j) \end{cases}. \tag{6}$$

Then, the $\delta_P$ can be redefined as:

$$\delta_P = \sum_{i,j} W_{ij} \|P \cdot f_i - P \cdot f_j\|^2. \tag{7}$$

Thus, NAP algorithm can be regarded as the following optimization problem:

$$PE = \underset{P}{argmin} \sum_{i,j} W_{ij} \| P \cdot f_i - P \cdot f_j \|^2. \tag{8}$$

The traditional method to solve Equation (8) is to transform it into solving eigenvectors of the following eigenproblem [21]:

$$F(W - diag(WI))F^{\mathrm{T}} v = \lambda v. \tag{9}$$

The matrix $P = I - \Omega = I - \sum_i \omega_i \omega_i^T$ with $\|\omega_i\| = 1$ and the $\omega_i = Fv_i$. $I$ is a length column vector of all ones. The eigenvector $v$ of $N$-dim obtained from Equation (6) is the related to the working conditions. Rank all eigenvalues and select the larger ones corresponding eigenvectors to estimate the nuisance subspace.

In this paper, the objective function Equation (8) is directly put into the CNN solved by gradient descent iteration. More details are shown in Section 3.

## 3. Proposed CNN-NAP Method

In Reference [25], 29 common features of the signal were firstly extracted; then, these features of the signal under different working conditions were projected by NAP; finally, a back propagation neural network (BPNN) was adopted for classification. The method in Reference [25] has achieved a good classification effect, but the structure is complex. In this paper, a more intuitive and concise method using NAP for fault diagnosis is presented.

The proposed CNN-NAP method is based on the combination of NAP and CNN, and the structure is shown in Figure 2. The proposed method cancels the step of extracting signal features and directly applies the CNN to the training of original signal classification. Nuisance attribute projection is seen as an FC layer added before softmax regression. In the back propagation, in addition to the cross entropy between softmax outputs and the labels, the objective function also add Equation (8) to minimize. The difference is that since NAP is extended to multiple faults, the value of $W$ here is:

$$W_{ij} = \begin{cases} 1 & workingcondition(f_i) \neq workingcondition(f_j) \\ 0 & workingcondition(f_i) = workingcondition(f_j) \\ 1 & faultpattern(x_i) = faultpattern(x_i) \\ 0 & faultpattern(x_i) \neq faultpattern(x_i) \end{cases}. \tag{10}$$

In addition, note that the projection matrix $P$ minimizing the nuisance attribute in the sub-space must satisfy the idempotent condition (i.e., $P^T P = P$). Thus, finally, the objective function is:

$$argmin(CE + PE + IC), \tag{11}$$

where $CE$ represents the cross entropy between softmax outputs and the labels, it is expressed by:

$$CE = -\sum y_i ln a_i, \tag{12}$$

where $y$ represents real value, $a$ represents the output value of softmax, and $i$ represents the node number of the fault category. $PE$ is the value obtained by Equation (8), $IC$ is the value of $\|P^T P - P\|$. In this way, by minimizing the objective function can not only train the network but also obtain a better projection matrix. In the next section, the structure of the model will be introduced according to the measured data.

## 4. Validation of CNN-NAP

In this section, CNN-NAP is applied to the bearing measured signal under various rotating speeds to verify the validity of CNN-NAP. The bearing fault simulation bench is shown in Figure 3, and the test bearing is cylindrical roller bearing of NU250EM whose parameters are shown in Table 1.

**Table 1.** Bearing parameters.

| Inside Diameter | Outside Diameter | Thickness | Pitch Diameter | No. of Balls |
|---|---|---|---|---|
| 25 mm | 52 mm | 15 mm | 44.2 mm | 13 |

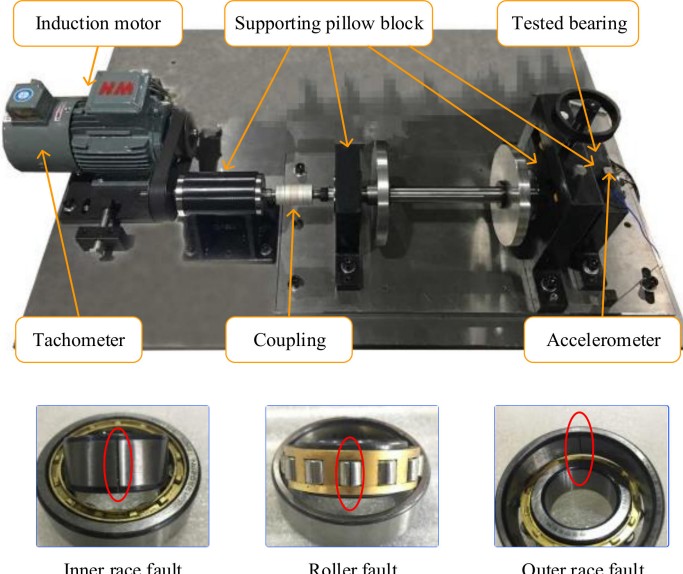

**Figure 3.** Bearing fault simulation experiment platform.

The bearing faults are artificially created by electron discharge machining technology and the crack depth is 0.5 mm on the outer-race, inner-race, and roller (OF, IF, and RF, respectively). The signals of five bearings health conditions (normal, OF, IF, RF and concurrent faults in the outer race and roller (ORF)) run under different rotating speeds are collected. The number of different bearing and roller (ORF)) run under different rotating speeds are collected. The number of different bearing signal sets under various rotating speeds and labels are shown in Table 2. Each sample contains 1200 data points, and the sampling frequency is 25.6 kHz.

**Table 2.** Bearing signal sets.

| Onehot Label | Fault Pattern | Training Data Sets | |
|---|---|---|---|
| (1,0,0,0,0) | Normal | RS (r/min) | 1000, 1300, 1500 |
| | | Sets | 100 |
| (0,1,0,0,0) | OF | RS (r/min) | 1000, 1300, 1500 |
| | | Sets | 100 |
| (0,0,1,0,0) | IF | RS (r/min) | 1000, 1300, 1500 |
| | | Sets | 100 |
| (0,0,0,1,0) | RF | RS (r/min) | 1000, 1300, 1500 |
| | | Sets | 100 |
| (0,0,0,0,1) | ORF | RS (r/min) | 1000, 1300, 1500 |
| | | Sets | 100 |

The testing data sets are time-varying signals under high-speed oscillation, and the rotation speed is irregularly changed in the interval of 0–1500 rpm, as shown in Figure 4. One hundred samples of each fault are randomly selected at three rotational speeds as training samples, respectively. Thus, there are 500 training samples, i.e., $n = 500$ in Figure 2. The input is a matrix of $500 \times 1200$; each row represents a sample with 1200 data points. In the designed CNN model, there are the three 'same' convolutional layers, three 'max' pooling layers and four fully connection layers. The size of the convolution kernel is $1 \times 10$, and the pooling window is $1 \times 10$. The activation function of convolution and connection layers is ReLU while C9 layer without it. It should be noticed that the features obtained

in the C9 layer need to go through the NAP projection. Other specific parameters are shown in Figure 2. In the process of back propagation, loss function is defined as Equation (11) to find the optimal weight and projection matrix $P$. Training epochs is 200 and training error is 0.001.

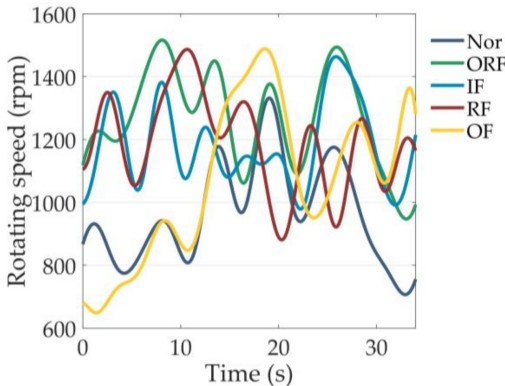

**Figure 4.** Testing datasets rotating speed.

In addition, 10,000 samples are randomly selected as testing data sets. The average training accuracy of 15 experiments is 100% and average testing accuracy of CNN-NAP is 99.57%. The specific results are shown in Figure 5. Taking inner-race fault as an example, Figure 6b shows the results of CNN-NAP learning in C9 layer, and it can be seen from the comparison Figure 6a that the features tend to be consistent. Therefore, this method can be used for rolling bearing fault diagnosis under various speeds conditions.

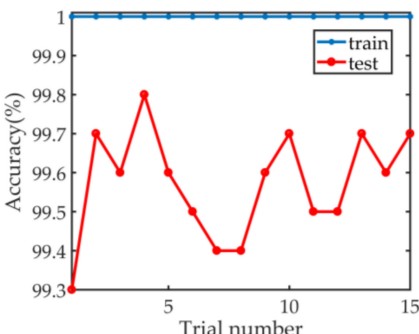

**Figure 5.** Results of 15 trials.

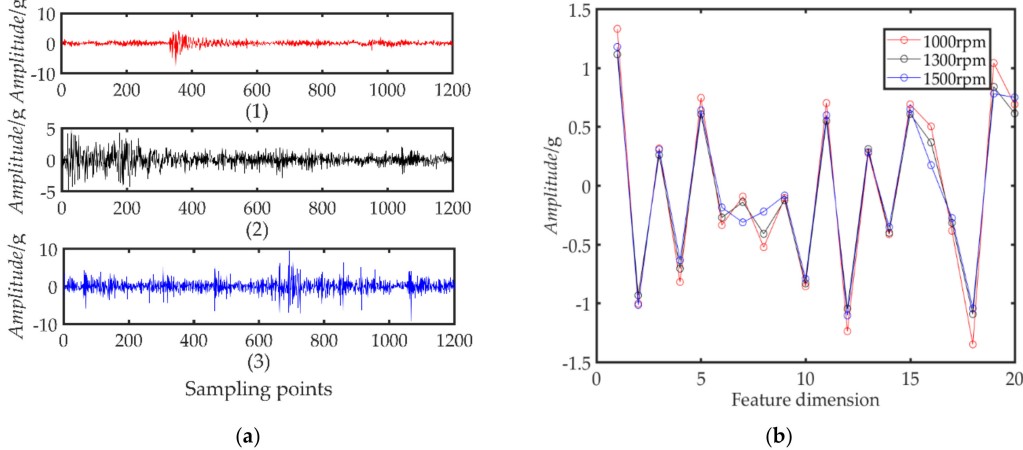

**Figure 6.** (**a**) samples of inner-race fault at three different speeds: (1) 1000 rpm; (2) 1300 rpm; (3) 1500 rpm; (**b**) the learned features of C9 layer.

## 5. Comparison Analysis

This chapter will discuss the advantages of the proposed method on the basis of proving the effectiveness of the proposed method, so as to provide some reference value for the subsequent research. The proposed method is compared and discussed with CNN and the method in Reference [25].

### 5.1. Compared with CNN

In order to prove the superiority of the proposed method over CNN, the C9 layer multiplication P operation is cancelled to train the same group of samples, i.e., training the same data only with CNN, the average classification accuracy of the 15 experiments is 93.9%. In order to illustrate the above argument more intuitively, t-SNE visualization method [32] is used to compare the data before and after P multiplication in C9 layer, as shown in Figure 7. t-SNE is a nonlinear dimension-reduction algorithm, which is very suitable for visualization of high-dimensional data from dimensionality reduction to 2D or 3D. As can be seen from Figure 7, in the case that there is no P multiplied in the C9 layer, there is a large gap between the same fault category. This is because the change of rotation speed interferes with the feature classification. However, under the projection of the trained matrix P on the C9 layer, the same kind of faults are clustered together, and the same fault points are not far away from each other. Therefore, NAP effectively eliminates the interference generated by the nuisance attribute (rotational speed) to the pattern recognition and improves the classification accuracy of CNN.

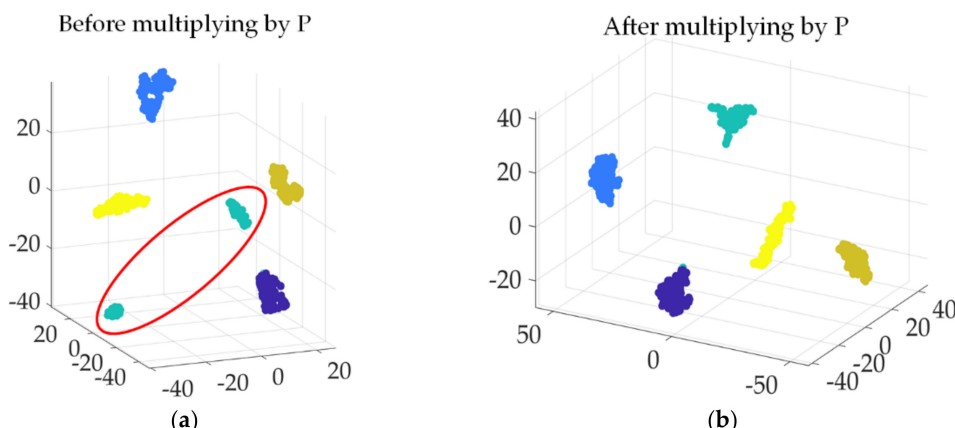

**Figure 7.** C9 layer tSNE visualization: (**a**) before multiplying by P and (**b**) after multiplying by P.

### 5.2. Compared with Previous Research

Reference [25] introduced NAP into the field of fault diagnosis for the first time and achieved good classification results, eliminating the interference caused by different speeds, loads, channels and noises when classifying. However, the steps of the proposed method are complex, and the structure is not simple enough. Compared with Reference [25], CNN-NAP has the following advantages (similar conclusions are presented in Figure 8):

1. CNN-NAP cancels the steps of extracting features manually and embeds the step of calculating projection matrix P directly into the training of neural network.
2. The original method to get the projection matrix is to transform the optimization problem into the problem of finding the eigenvalues and eigenvectors. However, the CNN-NAP method directly gets the optimization problem through the neural network training, and does not need to get the projection matrix separately.
3. CNN-NAP extended NAP to multiple fault cases, and the obtained projection matrix P could eliminate the nuisance attributes under each fault at the same time. In Reference [25], the corresponding number of projection matrixes P are needed for different fault types.

These three improvements greatly simplify the structure of the method and make the CNN-NAP much clearer. In addition, by training and testing the datasets in chapter 3 with the method mentioned in the Reference [25], the classification accuracy is 99.7%, which is almost the same as the CNN-NAP. Therefore, compared with the method in the Reference [25], CNN-NAP has a broader application prospect and a wider range of applications.

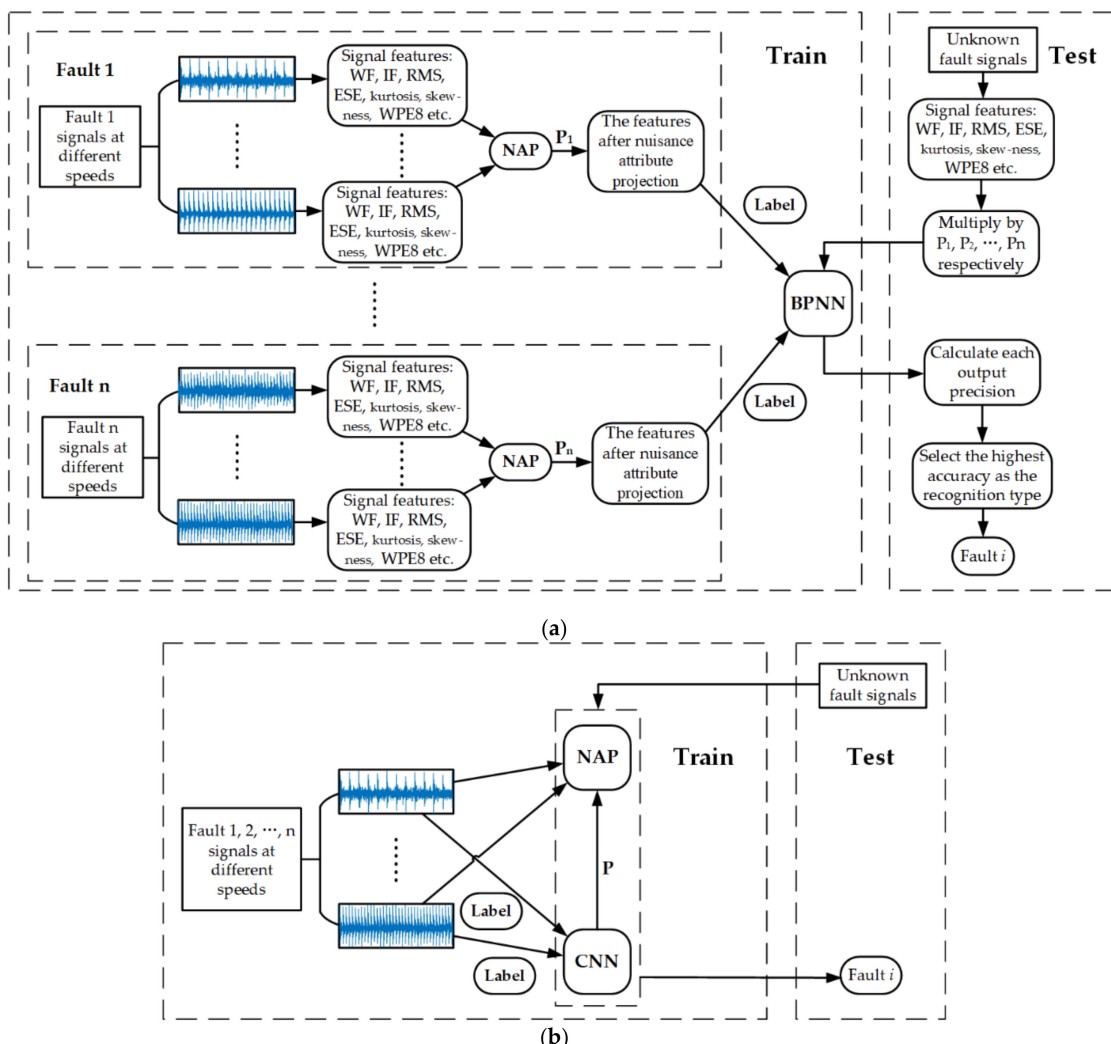

**Figure 8.** Fault diagnosis flowcharts: (**a**) the method in Reference [25]; (**b**) CNN-NAP.

Furthermore, it has been proved in Reference [25] that NAP is applicable to eliminate the interference properties including rotational speed, load, channel and so on. Limited by the experimental conditions, this paper did not discuss the elimination effect of interference attributes such as load and channel. However, according to Reference [25], it can be inferred that CNN-NAP is equally effective.

## 6. Conclusions

In this paper, a novel CNN-NAP method, which integrates NAP into CNN, was proposed for intelligent fault diagnosis of rotating machinery, so as to eliminate the nuisance attribute of speed influence. Through the case study of a bearing dataset, it was shown that the proposed method could adaptively eliminate interference from the original signal, successfully applied in the variable speed bearing fault diagnosis. In addition, the comparison analysis showed that the proposed method is better than CNN and the method in the Reference [25]. Compared with CNN, the proposed method improves the classification accuracy and makes the diagnosis results more reliable. Compared with



the method in the Reference [25], the proposed method has a more concise framework, especially when calculating the projection matrix of NAP, no matter how many fault types, only one projection matrix P is needed to be solved, which solved the problem of requiring multiple P in Reference [25]. The proposed method has great advantages over the non-resonance based approach. For example, the Auto Regression Moving Average method needs to apply the AIC criterion to determine the model's order, and then test the coefficient significance and the noise normality. The proposed method is much simpler and does not need to consider the problem of model's order and noise. It is directly driven by the sample data. In short, the proposed method can adaptively learn fault characteristics, making it more convenient for intelligent fault diagnosis and processing of big data. However, this method does not solve the high complexity of convolutional neural network itself. Therefore, developing a more powerful CNN is still a direction of future studies. In addition, this method should be extended to more related fields, such as earthquake engineering. Earthquake are divided into near field and far field with different frequency characteristics [33,34]. With enough training data, the proposed method should be able to eliminate the influence of nuisance attribute and open a new opportunity for ground motion characterization.

The ultimate purpose of mechanical fault diagnosis is to ensure the safe and stable operation of equipment. How to find more effective algorithms, conduct comprehensive analysis, determine the defects and faults of equipment, ensure the efficient operation of equipment, and create greater benefits for enterprises are the main directions of future research.

**Author Contributions:** Conceptualization, H.M. and Z.A.; methodology, H.M.; software, H.M. and Z.A.; validation, H.M. and S.L.; formal analysis, H.M. and S.L.; investigation, S.L.; resources, S.L.; data curation, Z.A.; writing—original draft preparation, H.M.; writing—review and editing, Z.A.; visualization, H.M.; supervision, S.L.; project administration, S.L.; funding acquisition, H.M., S.L and Z.A.

**Funding:** This research was funded by the National Natural Science Foundation of China, Grant No. 51675262, the National science and technology major projects, Grant No. 2017-IV-0008-0045 and the Advance research field fund project of China, Grant No. 6140210020102.

**Conflicts of Interest:** The authors declare no conflict of interest.

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
