# Peer review of "A Fault Diagnosis Approach for Rolling Bearing Based on Convolutional Neural Network and Nuisance Attribute Projection under Various Speed Conditions"

_applsci, doi:10.3390/app9081603_

Round 1

Reviewer 1 Report

See attached and apply the whole comments please.

Author Response

Thanks to the editor and reviewers for these suggestions. We have considered the comments from reviewers carefully and made revisions to address their concerns. Detailed response is given for each reviewer comment below and highlighted in the manuscript. We hope this revised version is acceptable for publication.

Specific/Technical Comments:

Point 1: The original contribution should somehow reflect its contribution to the relevant professional area. As such, the authors should put commentary about strengths and weaknesses of the proposed method in the conclusion part.

Response 1: Thanks to the reviewer for the pertinent suggestions. Since the strengths of proposed method have been described in the conclusion part, only the weaknesses are added: “However, this method does not solve the high complexity of convolutional neural network itself.” (Line 269-270)

Point 2: The title of sections 5.2 is not correctly presented. “Compared with ref 21” is not a professional way to address a section title. Alternatively, the authors should correctly find a relevant name for the section.

Response 2: Thanks to the reviewer’s comment. The title of sections 5.2 has been changed to Compared with previous research.” (Line 224)

Point 3: The introduction of the manuscript is not acceptable in the present form and should under major editions. The introduction should be re-constructed following the comment below:

Review the two following papers and address them in as supportive literature. Also, try to link your original contribution to the recent work mentioned:

i. Zhang, Y. and Randall, R.B., “Rolling element bearing fault diagnosis based on the combination of genetic algorithms and fast kurtogram”, Mechanical Systems and Signal Processing, 2009, 23(5), pp. 1509-1517.

ii. Wang, L., Liu, Z., Miao, Q. and Zhang, X., “Time–frequency analysis based on ensemble local mean decomposition and fast kurtogram for rotating machinery fault diagnosis”, Mechanical Systems and Signal Processing, 2018, 103, pp. 60-75.

Response 3: Thanks for this suggestion. In the introduction part, we have enriched the relevant literature according to your suggestions:

With the development of technologies of the sensor, signal processing and engineering measurement, many methods have been developed in recent years, such as wavelet analysis, empirical mode decomposition (EMD), fast kurtogram and so on [5,6]. However, these methods have some limitations in analyzing variable speed signals.” (Line 40-43)

We have added the references as follows:

5.            Lei, W.; Liu, Z.; Qiang, M.; Xin, Z. Time–frequency analysis based on ensemble local mean decomposition and fast kurtogram for rotating machinery fault diagnosis. Mechanical Systems & Signal Processing 2018, 103, 60-75.

6.            Zhang, Y.; Randall, R.B. Rolling element bearing fault diagnosis based on the combination of genetic algorithms and fast kurtogram. Mechanical Systems & Signal Processing 2009, 23, 1509-1517.

Point 4: In the reviewer’s opinion, the literature review is not acceptable in the present form and the following references should be added and reviewed. In the manuscript, the authors have proposed a new methodology; however, one important, as well as promising application is totally missing in the paper. The application of the proposed method in other fields is a very important component and should be reflected in your paper. Specifically, the proposed fault diagnosis can be widely used in random impulses in “earthquake engineering”. For the authors’ information, earthquake are mainly divided into two categories: near-field and far-field earthquakes. Each of them has their frequency content and have different effects on buildings. Your proposed framework can be applied for further fault diagnosis in ground motion record characterizations. As such, the authors should put couple of sentences in their introduction part, talking about the “possibility” of their method in being applied in “earthquake engineering”. The authors should mention that their work opens a new opportunity for ground motion characterization. Authors should use the following recent papers, as the references in earthquake engineering, and should address them in their text and list of references:

i. Rezaei Rad A, Banazadeh M. Probabilistic risk-based performance evaluation of seismically base-isolated steel structures subjected to far-field earthquakes. Buildings 2018;8. doi:10.3390/buildings8090128.

ii. Tajammolian H, Khoshnoudian F, Rezaei Rad A, Loghman V. Seismic Fragility Assessment of Asymmetric Structures Supported on TCFP Bearings Subjected to Near-field Earthquakes. Structures 2018;13:66–78. doi:10.1016/J.ISTRUC.2017.11.004.

Response 4: Thanks to the reviewer for the pertinent suggestions. We think it's very interesting and exciting to come up with the proposed method that can be used in other fields. Therefore, we have added:

“In addition, this method should be extended to more related fields, such as earthquake engineering. Earthquake are divided into near field and far field with different frequency characteristics [33,34]. With enough training data, the proposed method should be able to eliminate the influence of nuisance attribute and open a new opportunity for ground motion characterization.” (Line 271-275)

We have added the references as follows:

33.          Rezaei Rad, A.; Banazadeh, M. Probabilistic Risk-Based Performance Evaluation of Seismically Base-Isolated Steel Structures Subjected to Far-Field Earthquakes. Buildings 2018, 8.

34.          Tajammolian, H.; Khoshnoudian, F.; Rad, A.R.; Loghman, V. Seismic Fragility Assessment of Asymmetric Structures Supported on TCFP Bearings Subjected to Near-field Earthquakes. Structures 2018, 13, 66-78.

Point 5: What are the various sources of vibration in bearings targeted for recognition and fault attribution through the signal processing methods (the kurtogram approach)? You should explicitly mention that in your paper. You can read the following paper and refer to that more in detail.

i. Lynagh, N., Rahnejat, H., Ebrahimi, M. and Aini, R., “Bearing induced vibration in precision high speed routing spindles”, Int. J. Machine Tools and Manufacture, 2000, 40(4), pp. 561-577

Response 5: Thanks to the reviewer’s comment. We have done it:

Abnormal vibration signal will be generated due to the variable compliance effect, manufacturing defects and assembly related problems [3]. These signals can be processed to determine the source of fault, and then repaired and maintained accurately.

We have added the references as follows:

3.            Lynagh, N.; Rahnejat, H.; Ebrahimi, M.; Aini, R. Bearing induced vibration in precision high speed routing spindles. International Journal of Machine Tools & Manufacture 2000, 40, 561-577.

Point 6: Line 205 and 206, the sentence: “In order to prove the superiority of our method over CNN, the C9 layer multiplication P operation was cancelled to train the same group of samples, and the classification accuracy is 94.5%” is not clear. Expand it to make it more precise.

Response 6: Thanks for this suggestion. Cancelling the projection matrix P of C9 layer is equivalent to training the data only with convolutional neural network. We have added:

In order to prove the superiority of the proposed method over CNN, the C9 layer multiplication P operation is cancelled to train the same group of samples, i.e. training the same data only with CNN, the average classification accuracy of the 15 experiments is 93.9%.” (Line 212-214)

Point 7: In the reviewer’s opinion, the authors should demonstrate the advantage of their method over some non-resonance based approach to deal with bearing fault diagnosis, such as Auto Regression Moving Average method, in the conclusion part.

Response 7: Thanks to the reviewer’s comment. We have added:

The proposed method has great advantages over the non-resonance based approach. For example, the Auto Regression Moving Average method needs to apply AIC criterion to determine the model’s order, and then test the coefficient significance and the noise normality. The proposed method is much simpler and does not need to consider the problem of model’s order and noise. It is directly driven by the sample data. (Line 265-269)

Comments on Grammatical Issues and English Academic Writings:

Point 1: The manuscript does not completely follow the academic English writing rules and it requires English editions. It is recommended to go through the text again and the punctuation rules.

Response 1: We read the text again and expressed our ideas follow the academic English writing rules and corrected punctuations.

Point 2: The authors should be aware of using appropriate articles “the”, “an”, and “a”. Please review the text again and be sure you have used the appropriate structure. For instance, in line 214, “THE same fault points” should be used.

Response 2: Thanks to the reviewer’s comment. Articles have been added or corrected.

Point 3: Overall, you should keep a same verb tense for each section. The literature-review section in past, the methodology and research outcome in the present, and the conclusion in the past form.

Response 3: Thanks to the reviewer’s comment. Verb tenses have been corrected.

Point 4: It is recommended to stick to a same voice in the text. It should be either passive or active. Double check this in the manuscript. Avoid using “our study” “in our work”, etc.

Response 4: Thanks to the reviewer’s comment. All active phrases have been deleted.

Minor Comments:

Point 8: You should improve the quality of Figures 4, and 5.

Response 8: The quality of Figures 4 and 5 has been improved.

Point 9: Make the font size of Figure 2 larger.

Response 9: The font size of Figure 2 has become larger.

Point 10: Please note that the nomenclature of all the mathematical symbols should be reflected in your paper.

Response 10: The nomenclature of all mathematical symbols has been reflected in the paper.

Reviewer 2 Report

The paper presents a new methodology to rolling bearing fault diagnosis using CNN and NAP for various speed conditions.
The are some suggestions to improve the paper prior publication:
1. A few more papers need to be included in Introduction. The additional cited paper should described the method used for varying speed
   For example: Circular domain features based condition monitoring for low speed slewing bearing.
2. Please increase the resolution of Fig. 4 and 5.
3. The proposed method was compared to Ref. [21], is Ref. [21] used CNN as well? if not, then it is not apple to apple comparison. Then you need to compare with published paper that present the application of CNN in bearing fault diagnosis.

Author Response

Thanks to the editor and reviewers for these suggestions. We have considered the comments from reviewers carefully and made revisions to address their concerns. Detailed response is given for each reviewer comment below and highlighted in the manuscript. We hope this revised version is acceptable for publication.

Point 1: A few more papers need to be included in Introduction. The additional cited paper should described the method used for varying speed. For example: Circular domain features based condition monitoring for low speed slewing bearing.

Response 1: Thanks to the reviewer for the pertinent suggestions. We have added the references as follows:

2.            Caesarendra, W.; Kosasih, B.; Tieu, A.K.; Moodie, C.A.S. Circular domain features based condition monitoring for low speed slewing bearing. Mechanical Systems & Signal Processing 2014, 45, 114-138.

Point 2: Please increase the resolution of Fig. 4 and 5.

Response 2: Thanks to the reviewer’s comment. The quality of Figures 4 and 5 has been improved.

Point 3: The proposed method was compared to Ref. [21], is Ref. [21] used CNN as well? if not, then it is not apple to apple comparison. Then you need to compare with published paper that present the application of CNN in bearing fault diagnosis.

Response 3: Thanks to the reviewer’s comment. In sections 5, we have compared the proposed method with CNN, and the results show that the proposed method has better classification accuracy than CNN. Ref. [21] (now [25]) is the first introduction of NAP into fault diagnosis. Therefore, compared with Ref. [21], we mainly want to explain that our method applies NAP better in fault diagnosis and has a more concise framework.

Round 2

Reviewer 1 Report

The authors have addressed the comments.